

# Resampling-based methods for biologists

John R. Fieberg, Kelsey Vitense and Douglas H. Johnson

Department of Fisheries, Wildlife, and Conservation Biology, University of Minnesota, St. Paul, MN, USA

## ABSTRACT

Ecological data often violate common assumptions of traditional parametric statistics (e.g., that residuals are Normally distributed, have constant variance, and cases are independent). Modern statistical methods are well equipped to handle these complications, but they can be challenging for non-statisticians to understand and implement. Rather than default to increasingly complex statistical methods, resampling-based methods can sometimes provide an alternative method for performing statistical inference, while also facilitating a deeper understanding of foundational concepts in frequentist statistics (e.g., sampling distributions, confidence intervals, $p$-values). Using simple examples and case studies, we demonstrate how resampling-based methods can help elucidate core statistical concepts and provide alternative methods for tackling challenging problems across a broad range of ecological applications.

## INTRODUCTION

Falsifiable hypotheses and replication are two cornerstones of science (*Johnson, 2002*; *Popper, 2005*). Replication is also critical for understanding key statistical concepts in frequentist statistics (Box 1). Yet, as researchers, we typically encounter and analyze one data set at a time, making it difficult to understand the concept of a *sampling distribution* (i.e., the distribution of a sample statistic across repeated samples from the same population). Fully appreciating sampling distributions requires conceptualizing the process of repeatedly collecting new data sets, of the same size and using the same methods of data collection, and analyzing these data in the same way as in the original analysis. Similarly, *null distributions*, used to test statistical hypotheses and calculate $p$-values, require that we consider the distribution of our statistics (means, regression coefficients, etc.) across repeated hypothetical data collection and analysis efforts, while adding a further constraint that the null hypothesis is true.

Rather than serving as a unifying concept, the importance of variability across different replicate samples is often lost upon students when they take their first statistics course. The traditional formula-based approach to teaching gives the impression that statistics is little more than a set of recipes, each one suited to a different table setting of data. Most of these recipes rely on large-sample assumptions that allow us to derive an appropriate (albeit approximate) sampling distribution. For example, the sampling distribution of many statistics will be well approximated by a Normal distribution for large samples.

Corresponding author
John R. Fieberg, Jfieberg@umn.edu

**Box 1 Key concepts in frequentist statistics.**

- A *sampling distribution* is the distribution of sample statistics computed using different samples of the same size from the same population.
- A *bootstrap distribution* is a distribution of statistics computed using different samples of the same size from the same *estimated* population formed by merging many copies of the original sample data. Alternatively, the sample data may be used to estimate parameters of a statistical distribution, and then this distribution can be used to generate new samples. This alternative is termed the *parametric bootstrap*.
- A *null* or *randomization distribution* is a collection of statistics from samples simulated assuming the null hypothesis is true.
- The *standard error* of a statistic is the standard deviation of the sampling distribution. When forming confidence intervals, we can estimate the standard error using the standard deviation of a bootstrap distribution. When calculating $p$-values, we can estimate the standard error using the standard deviation of the randomization distribution.
- *2 SE rule*: when statistics have bell-shaped (i.e., approximately Normal) sampling distributions, we expect roughly 95% of sample statistics to be within 2 standard deviations of the mean of the sampling distribution.
- A *confidence interval* for a parameter is an interval computed from data using a method that will include the parameter value for a specified proportion of all samples (e.g., 95% of the time for a 95% confidence interval).
- The *p-value* is the chance of obtaining a sample statistic as extreme as (or more extreme than) the observed sample statistic, if the null hypothesis is true.

This result makes it possible to derive analytical formulas for calculating confidence intervals and $p$-values for simple one-sample problems (e.g., involving means or proportions) and two-sample problems (e.g., differences between group means or proportions). Additionally, sampling distributions of coefficients in linear regression models follow $t$-distributions when observations are independent and residuals are Normally distributed with constant variance. Introductory courses may also introduce $\chi^2$ and $F$ distributions, and mostly rely on analytical formulas for performing statistical inference.

Most problems facing ecologists are more complicated than the univariate problems explored in introductory statistics courses; further, assumptions of linear regression are often untenable. Each new challenge requires a new recipe, chosen from a more specialized and difficult-to-follow cookbook. Modern statistical methods, such as generalized linear models and random-effects models, allow one to relax assumptions that the residuals are Normally distributed, have constant variance, and that cases are independent. These methods are powerful, but to understand them fully requires a background in mathematical statistics, which most biologists lack. Consider, for example, generalized linear mixed models (GLMMs). These methods are widely available in modern statistical software, and biologists are routinely encouraged to analyze their data using these methods. Yet, we frequently encounter biologists who have fit GLMMs but do not know how to determine if the models are appropriate for their data. They struggle to describe fitted models using equations or text. And they frequently do not know how to create effect plots to visualize how the mean response changes with changes in predictor values, which for models with a non-linear link function requires integrating over the distribution of the random effects (*Fieberg et al., 2009*). Simply put, few biologists have taken the requisite coursework in calculus and mathematical statistics to understand these methods, and many are unfamiliar with common statistical terminology (e.g., expected value) or probability distributions other than the Normal distribution.

Resampling methods, including permutation procedures, often offer an attractive alternative to Normal-based inferential methods; they are adaptable and require fewer assumptions. Historically, the main limitations in applying these methods were that they were computationally intensive and often required custom-written computer code (*Cobb, 2007*). These limitations are no longer a significant concern except for extremely large data sets, given the availability of personal computers and open-source statistical software with packages for implementing resampling-based methods (e.g., *Davison & Hinkley, 1997*; *Pruim, Kaplan & Horton, 2017*; *Canty & Ripley, 2019*; *Simpson, 2019*; *Oksanen et al., 2019*; *R Core Team, 2019*).

Our objectives of this paper are to:

1. Illustrate how resampling-based approaches can facilitate a deeper understanding of core concepts in frequentist statistics (e.g., standard errors, confidence intervals, $p$-values).
2. Demonstrate through simple examples and case studies how resampling-based methods can provide solutions to a range of statistical inference problems.

For resampling-based methods to work appropriately, researchers need to be able to generate data sets that preserve the structure of the original data set (e.g., any clustering or other forms of correlation). This requirement can easily be meet with relatively simple computer code when addressing most univariate problems encountered in a first semester statistics course. The benefits of using resampling-based methods are often greatest, however, when analyzing complex, messy data. Determining appropriate solutions in these situations can be more challenging. Nonetheless, solutions are often available in open-source software and rely on the same set of core principles. Our case studies provide a few examples of the types of problems that can be addressed using resampling-based methods, but in truth they barely scratch the surface of what is possible. For a more in-depth treatment of bootstrapping and permutation tests, we refer the reader to *Davison & Hinkley (1997)* and *Manly (2006)*.

This article is written primarily for the applied biologist with a rudimentary understanding of introductory statistics, but we also expect it will be of interest to instructors of introductory statistics courses. In particular, our first objective can be seen as an argument for replacing traditional approaches to teaching introductory statistics with an approach that relies heavily on computational methods, an argument that is increasingly supported by data (*Tintle et al., 2011*, *2012*, *2015*; *Chance, Wong & Tintle, 2016*; *VanderStoep, Couch & Lenderink, 2018*). To that end, several educators have developed applets for explaining and understanding key concepts in frequentist statistics (e.g., http://www.rossmanchance.com/ISIapplets.html and http://www.lock5stat.com/StatKey/index.html). Here, we provide a tutorial review emphasizing the `mosaic` package in R, which was developed specifically to facilitate teaching resampling-based methods in introductory statistics courses (*Pruim, Kaplan & Horton, 2017*). We have chosen this approach because R has become a sort of lingua franca among ecologists. We begin by introducing key foundational concepts (sampling distributions, confidence intervals, null

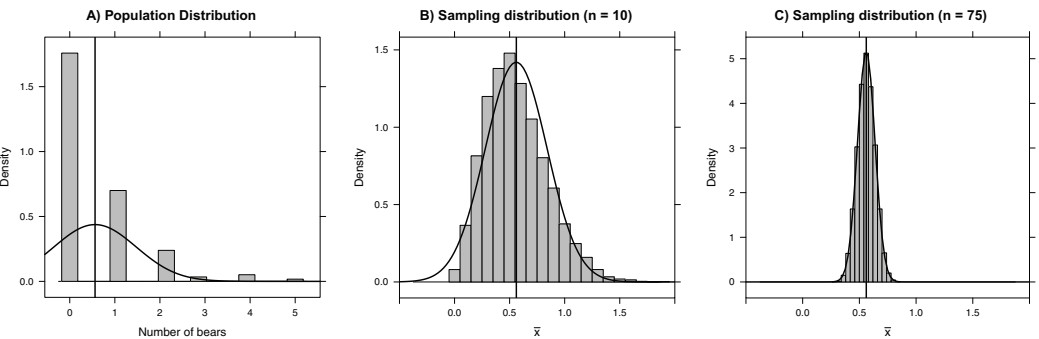

**Figure 1** Histograms depicting (A) the number of bears in each of 164 roughly 3 × 3 km quadrats on Rowley Island in northern Foxe Basin, Nunavut (*Stapleton et al., 2014*), (B) the sampling distribution of the mean number of bears in 10,000 simple random samples of size 10 plots, and (C) the sampling distribution of the mean number of bears in 10,000 simple random samples of size 75 plots. The vertical line gives the population mean and the smooth curves depict Normal approximations to the distributions in each panel.                     

distributions and *p*-values) using simple examples that can be analyzed with a few lines of code. We then consider a series of increasingly complex case studies that demonstrate how these concepts apply more broadly. To facilitate learning, we have archived all data and R code needed to re-create our examples at https://bootstrapping4biologists.netlify. com. In addition, the data and code have been curated and included in the Data Repository of the University of Minnesota (*Fieberg, Vitense & Johnson, 2020*).

## Understanding foundational concepts using simulation-based methods

### Sampling distributions

The sampling distribution of a statistic tells us about the values we might expect for the statistic if we were to repeatedly collect data sets of the same size from the same population and using the same sampling protocols. Again, this is a difficult concept to grasp because we usually only get to collect one data set. However, it is easy to use a computer to generate multiple samples from a population, and hence calculate multiple statistics to approximate the sampling distribution. To illustrate, we explore here the sampling distribution of a sample mean using a data set containing the count of the number of polar bears within each of 164 roughly 3 × 3 km quadrats on Rowley Island in northern Foxe Basin, Nunavut (Fig. 1A; *Stapleton et al., 2014*). First, we read in the data set which contains two variables, an id for each quadrat (`Quadrat`) and a count of the number of bears in each quadrat (`Num.Bears`), making use of the `here` package to make it easy to read in data from a subdirectory (*Müller, 2017*). We use <- to assign the data to an object called `bdat`. We then use the `head` function to look at the first five rows of the data set (here and throughout the paper, we provide R code followed by its associated output).

```
bdat <- read.csv(here("data", "bears.csv"))
head(bdat, n=5)
```

```
  Quadrat Num.Bears
1       1        2
2       2        1
3       3        0
4       4        0
5       5        1
```

We use the `mean` function to calculate the mean number of bears counted per quadrat in the population (i.e., the set of 164 quadrats). This function, like most in the `mosaic` package, follows a common syntax, which for statistical summaries of a single variable is: `goal(~variable name, data=Mydata)` (*Pruim, Kaplan & Horton, 2017*).

```
mean(~Num.Bears, data=bdat)
```

```
[1] 0.561
```

In this situation, we can calculate the population mean, $\mu = 0.561$, exactly because we have polar bear counts for all sample units. Nonetheless, we will use these data to illustrate the concept of the sampling distribution. Specifically, we consider a hypothetical situation in which we have access to the counts of bears in each of 164 quadrats (i.e., the population), to look at what *could* have happened if we had only sampled a random subset of 75 quadrats. Our sample statistic, the sample mean ($\bar{x}$), provides an estimate of the population mean ($\mu$) bear count per quadrat. Below, we repeatedly take simple random samples of 75 quadrats using the `sample` function and calculate the sample mean for each of the resulting data sets. We store the resulting sample means in an object named `samp.mean`. This feat can be accomplished with essentially one line of code using the `do` function in the `mosaic` package, where its argument (10,000 in this case) tells R how many times to repeatedly execute the code within the { } (To reproduce our results exactly, readers should set the seed of the random number generator using `set.seed(03222007)`):

```
samp.mean<-do(10000)*{
  mean(~Num.Bears, data=sample(bdat, size= 75))
}
```

The `do` function captures the 10,000 sample means and stores them in a variable named `result` within the `samp.mean` object. We can look at the first five sample means using the `head` function in R:

```
head(samp.mean, n=5)
  result
1   0.507
2   0.600
3   0.453
4   0.480
5   0.520
```

We can easily visualize the sampling distribution using a histogram or density plot (Fig. 1C), and we can calculate the standard error using the `sd` function in the `mosaic`
library (the standard error is just the standard deviation of the sampling distribution; Box 1):

```
(se.mean<-sd(~result, data=samp.mean))
```

```
[1] 0.0772
```

When statistics have bell-shaped (i.e., approximately Normal) sampling distributions, we expect 95% of sample statistics to be within roughly 2 standard deviations of the mean of the sampling distribution (1.96 standard deviations to be more exact); we will refer to this result as the **2 SE rule** (Box 1). We can use the above simulation results (i.e., the 10,000 sample means contained in `samp.mean`), to verify this claim. First, we find this interval by taking the mean of the sampling distribution ± 2 times the SE of the mean (we stored this SE in an object called `se.mean` and use the `c` operator to create a vector containing the values −2 and 2).

```
(rule.2SE<-mean(~result, data=samp.mean)+c(−2, 2)*se.mean)
```

```
[1] 0.407 0.715
```

We then determine the proportion of sample means that fall in this interval (i.e., the proportion of sample means that are greater than the lower threshold and less than the upper threshold) using the `tally` function. We use `I(result>= 0.407 & result <=0.715)` to create an indicator variable that is 1 when result is in the interval and zero otherwise.

```
tally(~I(result >= 0.407 & result <=0.715),
  data=samp.mean, format="proportion")
```

```
I(result >= 0.407 & result <= 0.715)
   TRUE     FALSE
0.9548   0.0452
```

We see that roughly 95% of the sample means fall within 2 standard errors of the mean.

## Central limit theorem

The sampling distribution of means (or sums) approaches that of a Normal distribution as the sample size increases. This result, given by the Central Limit Theorem, forms the basis of many formula-based methods of statistical inference and can be illustrated through simulation (*Kwak & Kim, 2017*). For example, the sampling distribution in our bear example is right-skewed for small sample sizes (e.g., $n = 10$; Fig. 1B); however, for samples of size 75, the sampling distribution is bell-shaped and well-approximated by a Normal distribution. Thus, rather than blindly trusting that the Central Limit Theorem applies, we can demonstrate it first hand by sampling repeatedly using different sample sizes. Additionally, the Central Limit Theorem guarantees for sufficiently large samples that the sampling distribution of $\bar{x}$ will be approximately Normal with 95% of the sample means falling within 2 standard errors of $\mu$ in the polar bear example. This result holds even though the *population distribution* is right-skewed and highly discrete, with individual bear counts taking on only integer values. Further, many estimators (including all maximum likelihood estimators) are calculated via a sum of independent measurements,

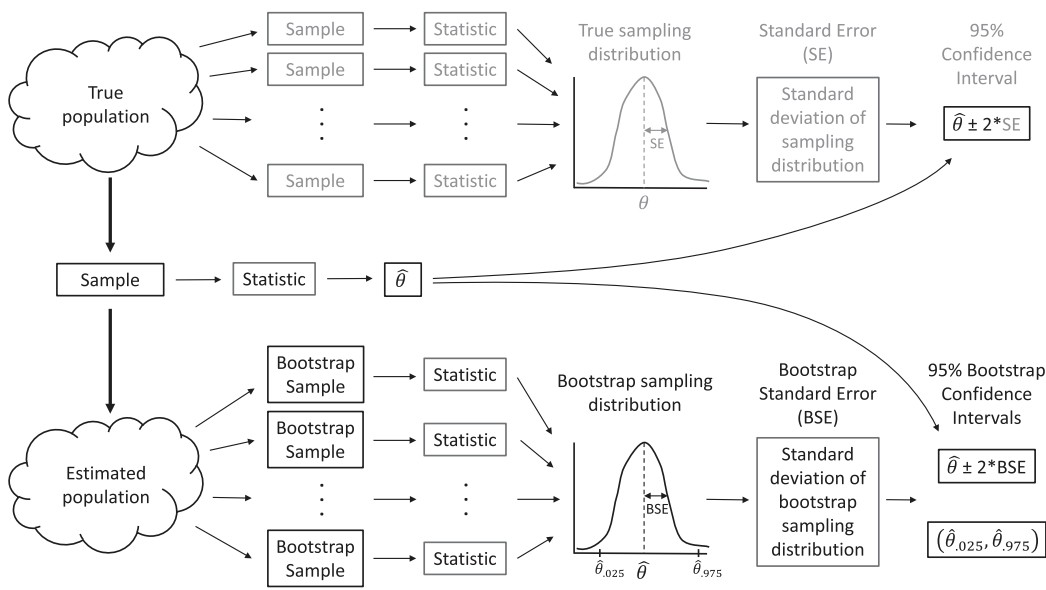

**Figure 2** A sampling distribution is the distribution of sample statistics computed using different samples of the same size from the same population. We can estimate characteristics of the sampling distribution (e.g., its standard deviation) using bootstrapping, in which we repeatedly sample from an estimated population. Each bootstrap sample should be the same size as the original sample, and bootstrap samples should be formed in such a way that they preserve the structure of the original data set (e.g., any clustering or other forms of correlation).

which ensures that their sampling distributions are asymptotically Normal. Understanding this powerful result helps uncover why many back-of-the-envelope calculations use *estimate* ± 2SE to form approximate 95% confidence intervals.

## Bootstrap confidence intervals

In real applications, we do not have access to data from the full population. Instead, we have but a single sample. When making inference to the population, via traditional parametric methods or with resampling-based methods, we must assume our sample data are representative of the population, as is assumed when a large number of observations are selected by simple random sampling. Then, we can use the distribution of values in our sample to approximate the distribution of values in the population. For example, we can make many, many copies of our sample data and use the resulting data set as an estimate of the whole population. With this estimated population in place, we could repeatedly sample from it, forming many data sets that are the same size as our original data set, and calculate statistics for each sample to approximate sampling distributions (Fig. 2). In practice, we do not actually need to make multiple copies of our sample data to estimate the population; instead, we form new data sets that are the same size as our original data set by sampling our original data with replacement, which effectively does the same thing. Sampling with replacement means that we select cases one at a time, and after each selection, we put the selected case back in the population so it can be chosen again. Thus, each observation in the original data set can occur zero, one, two, or more times in the generated data set, whereas it occurred exactly once in the original data set.
This process allows us to create a bootstrap sampling distribution to determine how much our estimates vary among repeated samples. We quantify this variability using the standard deviation of our estimates across repeated samples and refer to this standard deviation as our bootstrap standard error (BSE; Fig. 2). If the bootstrap sampling distribution is bell-shaped and centered on our sample statistic (indicating there is no sampling bias), then we can use this estimated standard error and the 2 SE rule for Normally distributed data to calculate a 95% confidence interval for our parameter of interest: estimate ± 2BSE. We can also think about repeating this process many times (i.e., collect data, estimate a parameter, use the bootstrap to calculate a SE and confidence interval), in which case we expect 95% of our confidence intervals to include the true population parameter of interest. That is, the bootstrap method for calculating a 95% confidence interval is what statisticians call well calibrated. In practice, we only have one data set, leading to a single confidence interval that either does or does not contain the true value of the population parameter, but knowing the method is well calibrated allows us to state that we are 95% *confident that our interval contains the population parameter*.

We illustrate this approach by considering data collected by the Minnesota Department of Natural Resources (MN DNR) to explore the potential impact of changing fishing regulations on the size distribution of northern pike (*Esox lucius*) in Medicine Lake, an approximately 460-acre lake in Beltrami County, MN. In 1989, the MN DNR instituted a slot limit of 22–30 inches in this lake (i.e., all caught fish within this size interval had to be released). We consider length data from 73 and 81 fish collected in trap nets in 1988 and 1993, respectively (before and after the fishing regulation was put in place). Importantly, these data come from only one lake, and many other factors may have changed between 1988 and 1993. Therefore, we must be cautious when interpreting any changes in the distribution of fish sizes (i.e., attributing the cause of length changes to the management regulation or generalizing results to other lakes). Nonetheless, we can ask, "How much did fish length, on average, change between 1988 and 1993 in Medicine Lake?" To address this question, we estimate the difference in the mean length of fish in the two samples. We also quantify our uncertainty in this estimated difference in means, recognizing that we would get a different estimate if we could go back and collect other samples of fish in those 2 years.

We begin by calculating the sample size, mean length in the 2 years, and the difference in sample means using the `tally`, `mean`, and `diffmean` functions in the `mosaic` library. Again, note that the functions in the `mosaic` library have a common syntax for data summaries involving two variables: `goal(y~x, data=Mydata)`. For example, `mean(y~x, data=Mydata)` will calculate the mean of y for each level of a categorical variable x, and `diffmean(y~x, data=MyData)` will calculate the difference in sample means when x is a categorical variable taking on only two levels. Here, we store the difference in means in an object named `effect_size`:

```
pikedat<-read.csv(here("data", "Pikedata.csv"))
tally(~year, data=pikedat)
```

```
year
1988 1993
  73 81
```

```
mean(length.inches~year, data=pikedat)
```

```
1988 1993
18.6 21.2
```

```
(effect_size <- diffmean(length.inches~year, data=pikedat))
```

```
diffmean
    2.58
```

We estimate that the mean size of pike increased by roughly 2.6 inches between 1988 and 1993. To evaluate uncertainty in our estimated effect size, we explore the variability in the difference in means across bootstrapped samples. Here, we use the sample data from 1988 and 1993 as our estimate of the distribution of lengths in the population in 1988 and 1993, respectively. We repeatedly sample (10,000 times) from these estimated populations and calculate our sample statistic (the difference in means) for each of these bootstrapped data sets, making use of the `do` and `resample` functions in the `mosaic` library. Whereas the `sample` function, by default, samples without replacement, the `resample` function samples with replacement; the `group` argument is used to ensure that bootstrapping is conducted separately for each `year`.

```
bootdist<-do(10000)*{
   diffmean(length.inches~year, data=resample(pikedat, group=year))
   }
```

Above, we stored the 10,000 differences in means as a variable named `diffmean` contained in the `bootdist` object that we created. We can look at the first five differences in means using the `head` function.

```
head(bootdist, n=5)
```

```
  diffmean
1     3.27
2     1.19
3     2.32
4     3.11
5     2.92
```

Because the bootstrap distribution is bell-shaped and centered on our sample statistic (Fig. 3), we can estimate a confidence interval for the difference in mean length in 1993 relative to 1988 using the 2 SE rule (i.e., based on an unstated Normal distributional assumption):

```
(SE<-sd(~diffmean, data=bootdist))
```

```
[1] 0.673
```

```
(CIa<-effect_size + c(-2, 2)*SE)
```

```
[1] 1.24 3.93
```

Alternatively, we could use quantiles of the bootstrap distribution, which does not require Normality (Fig. 2). The 0.025 quantile (or, equivalently, 2.5th percentile) refers to the value of $x$, such that 2.5% of the distribution falls below $x$. Here, we use the `qdata` function in the `mosaic` library to determine the 0.025 and 0.975 quantiles. These quantiles capture the middle 95% of the bootstrap distribution for the difference in means:

```
(CIb<-qdata(~diffmean, data=bootdist, p=c(0.025, 0.975)))
```

```
      quantile     p
2.5%      1.26 0.025
97.5%     3.90 0.975
```

For bell-shaped bootstrap distributions, Normal-based and percentile-based bootstrap confidence intervals work well and will usually give similar results. For small samples or skewed distributions, better methods exist (*Davison & Hinkley, 1997*; *Hesterberg, 2015*; *Puth, Neuhäuser & Ruxton, 2015*). Nonetheless, simple examples like this can facilitate understanding confidence intervals and other measures of uncertainty.

## Null distributions and *p*-values

Although null hypothesis testing has largely fallen out of favor in many disciplines, including ecology (*Johnson, 1999*; *Hobbs & Hilborn, 2006*), *p*-values, when fully understood, can play an important role in assessing the plausibility of statistical hypotheses (*De Valpine, 2014*; *Murtaugh, 2014*; *Dushoff, Kain & Bolker, 2019*). Further, it is critically important for biologists to be able to interpret the *p*-values they see in papers and in output generated by statistical software. To understand *p*-values, we must consider the sampling distribution of our statistic across repeated samples in the case where the null hypothesis is true. Simulation-based methods are also useful for understanding this concept, provided we can generate data sets consistent with the null hypothesis.

To illustrate, we will consider data from an experiment used to test various hypotheses about the mating preferences of female sagebrush crickets (*Cyphoderris strepitans*) (*Johnson, Ivy & Sakaluk, 1999*). Males of this species will often allow a female to eat part of their hind wings when mating, which decreases the male's attractiveness to future potential mates. To explore how diet might influence mating behaviors of females, *Johnson, Ivy & Sakaluk (1999)* randomly assigned 24 female crickets to either a low-nutrient (`starved`) or high-nutrient (`fed`) diet before putting them in a cage with a male cricket to mate. They then measured the waiting time before mating occurred. These data are contained in the `abd` package and were used by *Whitlock & Schluter (2009)* to introduce the Mann–Whitney *U*-test.

We can access the data using the `data` function after first loading the `abd` package:

```
library(abd)
data("SagebrushCrickets")
```

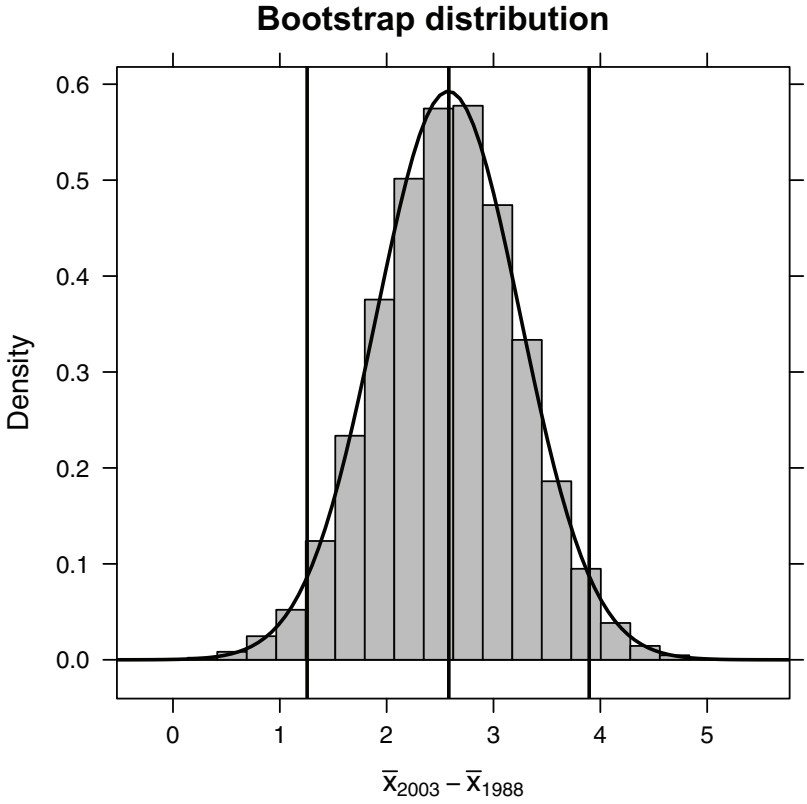

**Figure 3 Bootstrap distribution of the difference in mean length of fish (year 1993 relative to year 1988).** Outermost vertical lines indicate the 95% confidence interval using the percentile-based method, and the smooth curve illustrates the best-fit Normal distribution.

We then use the `str` function to explore the structure of the data set, finding that there are two variables: `treatment`, a factor variable with values `fed` or `starved`, and `time.to.mating`, a numerical variable containing the mating times for each cricket.

```
str(SagebrushCrickets)
```

```
'data.frame':       24 obs. of 2 variables:
$ treatment      : Factor w/ 2 levels "fed", "starved": 2 2 2 2 2 2 2 2 2 2 ...
$ time.to.mating: num 1.9 2.1 3.8 9 9.6 13 14.7 17.9 21.7 29 ...
```

Our objective is to test whether the mean waiting times depend on the diet of the female: $H_0 : \mu_{\text{fed}} = \mu_{\text{starved}}$ versus $H_A : \mu_{\text{fed}} \neq \mu_{\text{starved}}$ where $H_0$ and $H_A$ stand for the null and alternative hypotheses, and $\mu_{\text{fed}}$ and $\mu_{\text{starved}}$ represent the mean waiting times of crickets fed the high- and low-nutrient diets, respectively. It is natural to consider using the difference in sample means, $\bar{x}_{\text{fed}} - \bar{x}_{\text{starved}}$ to conduct our hypothesis test. Given the small sample sizes (we have 13 and 11 cases in the `fed` and `starved` treatment groups, respectively), we cannot safely assume the Central Limit Theorem ensures that the sampling distribution of $\bar{x}_{\text{fed}} - \bar{x}_{\text{starved}}$ is Normal. Further, the distribution of waiting times within each treatment group is far from Normal, and the measurements do not appear to

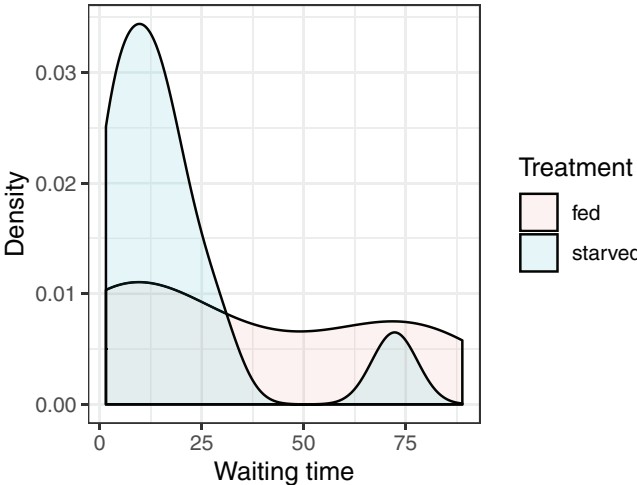

**Figure 4 Distribution of waiting times to mating for female sagebrush (*Cyphoderris strepitans*) crickets randomly assigned to either a low-nutrient (`starved`) or high-nutrient (`fed`) diet.**

be equally variable within each treatment group (Fig. 4). Thus, the assumptions of a parametric *t*-test are likely not met.

Although we could use a rank-based non-parametric test, such as the Mann–Whitney *U*-test (*Whitlock & Schluter, 2009*), we can gain additional insights into key statistical concepts (null distributions and *p*-values) if we construct our own randomization test, which also allows us to relax the Normality assumption.

We begin by estimating the mean waiting times for the two treatment groups (`fed` and `starved`) as well as the difference in sample means.

```
mean(time.to.mating~treatment, data=SagebrushCrickets)
  fed starved
  36.0   17.7
```

```
(effect_size2<-diffmean(time.to.mating~treatment,
data=SagebrushCrickets))
```

```
diffmean
  −18.3
```

We see that the mean time to waiting is 18.3 h longer for the crickets in the `fed` treatment group. A skeptic might point out that each time you repeat this experiment you will get a different value for this statistic (i.e., the difference in sample means). Perhaps the longer mean waiting time was just a fluke. Could that be the case? To find out, we need to determine what values we might expect to see for the difference in means when the null hypothesis is true. If the null hypothesis is true, then the labels (`fed` and `starved`) should not matter. So, to simulate a sample statistic (difference in means) that we might get if the null hypothesis were true, we can: (1) randomly shuffle the treatment variable among cases to form a new data set, and (2) calculate the difference in sample
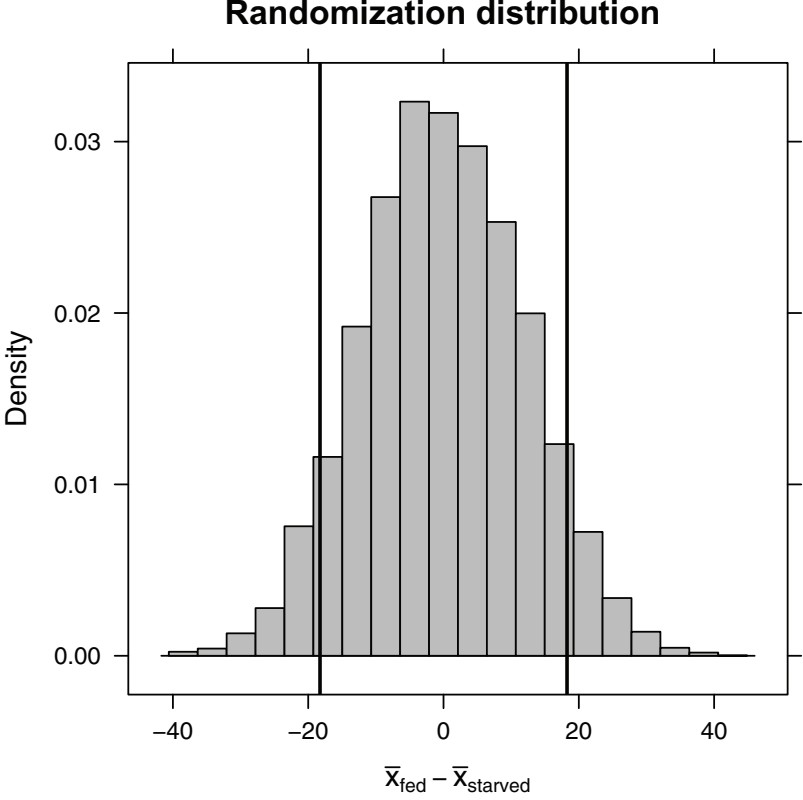

**Figure 5** Randomization distribution for the difference in mean waiting time until mating formed by shuffling treatment assignments (`fed` versus `starved`) among cases. This distribution informs us of the types of statistics (difference in means) that we would expect to see if the null hypothesis were true. Our observed sample statistic ($\bar{x}_{starved} - \bar{x}_{fed} = -18.3$, is indicated by the leftmost vertical line. When the null hypothesis is true, we would expect to get a difference in sample means $\leq -18.3$ or $\geq 18.3$ 13% of the time (i.e., the $p$-value is 0.13).

means. Using the `do`, `diffmean`, and `shuffle` functions in the `mosaic` library, we can repeat this process many times:

```
randomization_dist<-do(10000)*{
    diffmean(time.to.mating~shuffle(treatment), data=SagebrushCrickets)
}
```

Here, the `shuffle` function permutes the `treatment` variable, ensuring that, on average, the difference in sample means between the two treatment groups is equal to zero. Again, the do function stores the 10,000 differences in means in a variable named `diffmean` in the `randomization_dist` object that we created. Plotting this randomization distribution (Fig. 5), we find the range of statistics (difference in sample means) that we would expect to see if the null hypothesis were true. We can then use this information to determine the probability of observing an absolute difference in sample means as large or larger than we saw in our experiment, $|\bar{x}_{starved} - \bar{x}_{fed}| = 18.3$, if the null hypothesis is true. This probability is our two-sided $p$-value. We can approximate it using our randomization distribution by counting the number of times the difference in

sample means was ≤−18.3 or ≥18.3 (equivalent to the number of times the absolute value of the difference was ≥ 18.3).

```
(p.value<- tally(~abs(diffmean)>=18.3, data=randomization_dist,
                 format="proportion"))
```

```
abs(diffmean) >= 18.3
  TRUE FALSE
0.126  0.874
```

Given a $p$-value of 0.126, we expect that roughly 13% of the time we would get an absolute difference in mean waiting times as large or larger than 18.3 if the null hypothesis were true. Thus, our result is perhaps not all that surprising. Still, our sample sizes were small, leading to differences in sample means that were highly variable from sample to sample. It would behoove us to also report a confidence interval to go with our result, so the reader has an understanding of the range of effect sizes that are plausibly supported by the data. Using a bootstrap, we find that we are 95% confident that the difference in population means is between −38.9 and 3.6.

```
boot_dist<-do(10000)*{
   diffmean(time.to.mating~treatment, data=resample(SagebrushCrickets))
}
qdata(~diffmean, data=boot_dist, p=c(0.025, 0.975))
```

```
   quantile        p
2.5%   −38.91 0.025
97.5%    3.61 0.975
```

## Generalizations to other sample statistics

The methods illustrated here easily generalize to other statistics and data sets, provided cases can be assumed to be independent (e.g., there are no repeated measures on the same sample unit). For example, we could calculate a bootstrap distribution, and hence confidence interval, for a correlation coefficient using:

```
do(10000)*{cor(y~x, data=resample(Mydata))}
```

We could determine an appropriate null distribution for testing whether a set of regression coefficients are all zero versus an alternative hypothesis that at least one is non-zero by shuffling the response variable:

```
do(10000)*{lm(shuffle(y)~x1 + x2 + x3, data=Mydata)}
```

Shuffling the response variable breaks any association between $y$ and the predictor variables, thus meeting the assumption of the null hypothesis. It is tempting to think that we could also determine an appropriate null distribution for testing whether a regression coefficient for $x_2$ is zero, while adjusting for two other variables ($x_1$ and $x_3$) using:

```
do(10000)*{lm(y~x1 + shuffle(x2) + x3, data=Mydata)}
```

Although shuffling the values of $x_2$ for each case will break any relationship between $x_2$ and $y$, this process also breaks any association between $x_2$ and the other two predictor variables, $x_1$ and $x_3$. Thus, shuffling $x_2$ imposes additional unintended restrictions on the data beyond those required to ensure the null hypothesis is true. Alternative solutions have been proposed, which involve permuting residuals, either residuals of the response (*Freedman & Lane, 1983*; *Anderson & Robinson, 2001*), or the predictor at stake (*Collins, 1987*; *Dekker, Krackhardt & Snijders, 2007*). The latter approach is perhaps simplest, has good statistical properties, and is described below (see Appendix A for a simple coded example):

1. Fit a linear regression model relating $x_2$ to the other predictor variables in the model: `lm(x2~x1+x3, data=Mydata)`. The residuals from this model, which we label `x2r`, capture the part of $x_2$ that is not explained by the linear effect of other variables in the model (here, $x_1$ and $x_3$). If we replace `x2` in our original model with `x2r`, fitting the following model: `lm(y~x1+x2r+x3, data=Mydata)`, the coefficient, standard error, $t$-statistic, and $p$-value for `x2r` will be identical to the coefficient for `x2` in our original model.

2. Create an appropriate null distribution of $t$-statistics associated with the coefficient for `x2r` by shuffling `x2r`: `do(10000)*{lm(y~x1 + shuffle(x2r) + x3, data=Mydata)}`. The test is based on the $t$-statistic from the above randomization distribution rather than the coefficient for `x2r` itself because the $t$-statistic ensures the permutation distribution does not depend on additional unknown parameters, whereas the regression coefficient does not have this property.

As the above example illustrates, solutions to more complex applications may require custom-written code (and additional care), but the underlying concepts remain the same. For confidence intervals, we need to generate bootstrapped data sets that mimic how we obtained the original sample data, calculate our sample statistic for each of these data sets, and then use the variability in those statistics to form our confidence interval. For testing null hypotheses, we need to generate data sets for which we have assured compliance with the null hypothesis and then compare our observed statistic with those simulated under that null hypothesis. In the words of *Cobb (2007)*, "I like to think of (the core logic of statistical inference) as three Rs: randomize, repeat, reject. Randomize data production; repeat by simulation to see what's typical and what's not; reject any model that puts your data in its tail."

For computationally intensive applications, it may not be possible to generate as many as 10,000 random data sets as in our examples. In such cases, it is customary and important to include the original sample statistic in the permutation or bootstrap distribution (*Davison & Hinkley, 1997*; *Phipson & Smyth, 2010*). Doing so ensures statistical hypothesis tests are exact even for small numbers of permutations (*Phipson & Smyth, 2010*), that is, across repeated tests with a significance level of $\alpha = 0.05$, we will reject at most 5% of the time when the null hypothesis is true.
## Bootstrapping versus permutations

Our simple examples above demonstrate how to form confidence intervals by bootstrapping and how to test statistical hypotheses using permutations. In truth, one can use bootstrapping to perform hypothesis tests and permutations to form confidence intervals (e.g., by "inverting" a hypothesis test so that the confidence interval includes only parameter values for which we would fail to reject the null hypothesis). Ideally, the chosen approach (resampling or permutation) should reflect the original study design. For randomized experiments, it makes sense to use permutations that reflect alternative outcomes of the randomization procedure. By contrast, bootstrap resampling may be more appropriate for observational data where the "randomness" is reflected by the sampling procedure (*Lock et al., 2013*).

Consider again the fish length example used to demonstrate bootstrap confidence intervals. If we wanted to conduct a statistical hypothesis test that mean fish length was the same in both years, we could make the null hypothesis true (by adding the difference between the two sample means to the length of fish in the first sample), then bootstrap. This approach is illustrated below. We begin by creating a new variable, `length.null`, by adding the difference in sample means to all fish lengths collected in 1988. We can accomplish this task using the `mutate` function in the `dplyr` library (*Wickham et al., 2019*). The `ifelse` function evaluates the first expression (`year=="1988"`) and executes the next argument when true and the last argument when false. The difference in the mean of this new variable is zero (reflecting our null hypothesis).

```
delta<-diffmean(length.inches~year, data=pikedat)
pikedat<-mutate(pikedat,
   length.null=ifelse(year=="1988", length.inches+delta,
                      length.inches))
diffmean(length.null~year, data=pikedat)

diffmean
      0
```

We can then use the bootstrap to quantify variability in difference in sample means *when the null hypothesis is true*, from which we can calculate the $p$-value for our hypothesis test.

```
randfish<-do(10000)*{
   diffmean(length.null~year, data=resample(pikedat, group=year))
}
(p.value2<-tally(~abs(diffmean)>=delta, data=randfish,
format="proportion"))

abs(diffmean) >= delta
   TRUE FALSE
      0     1
```

None of the bootstrap differences in sample means was as large or larger than the difference we observed using our original data, resulting in a $p$-value $\approx 0$.

This example raises another important point of consideration when choosing an appropriate method for testing null hypotheses, namely differences between bootstrap- and permutation-based null distributions. In the sagebrush cricket example, we permuted treatment labels to create our null distribution for the difference in sample means. By permuting treatment labels, we ensure the *distribution* of the response variable, *y*, is the same for both treatment groups. Thus, means, variances, and higher-order moments will all, on average, be the same for the two groups. By contrast, the bootstrap-based test ensures only that the sample means of the two groups are, on average, the same under the null hypothesis. Both methods should reject the null hypothesis in cases where the population means differ, provided our sample size is large enough. However, the permutation-based test may also reject the null hypothesis (with probability > α) if the population means are the same in the two groups, but the population variances differ. Thus, the bootstrap-based test may be preferred if the goal is to detect only differences in population means, allowing for the possibility that the population variances differ.

## CASE STUDIES

We next consider a series of three case studies that demonstrate how the bootstrap can be used to calculate confidence intervals when faced with more challenging data-analytic problems. Specifically we show how to use the bootstrap for more robust inference in the face of assumption violations (Case Study 1), to estimate complex quantities that combine results from multiple analyses (Case Study 2), and to quantify model-selection uncertainty (Case Study 3). The data and R code needed to reproduce these examples are available at https://bootstrapping4biologists.netlify.com. In addition, we use the site to host the output that results from running all of the coded examples.

### Case Study I: relaxing the assumptions of linear regression

*Zuur et al. (2009)* lead off their popular book, *Mixed Effects Models and Extensions in Ecology with R*, with the statement (p. 19), "In Ecology, the data are seldom modeled adequately by linear regression models. If they are, you are lucky." Here, we consider one of the data sets from their book, containing measurements of species richness at 45 sampling stations in the Netherlands (five stations located within each of nine beaches, *Janssen & Mulder, 2004*, *2005*). When plotting species richness against an index of exposure of each beach to waves and surf, we see that mean species richness decreases linearly with exposure level (Fig. 6A), but the distribution of residuals is right skewed and, thus, non-Normal (Fig. 6B). In addition, observations were clustered in space so we might expect two observations from the same beach to be more alike than two observations from two different beaches.

The rest of Zuur et al.'s book is devoted to describing regression methods that allow one to relax these assumptions; for example, using random effects to account for correlation and different statistical distributions to relax the Normality and constant variance assumptions. Without knowledge of these tools, however, is there a way to estimate uncertainty in the regression line as a summary of the relationship between species richness and exposure (over the observed range of the data)? Yes! We just need to adapt

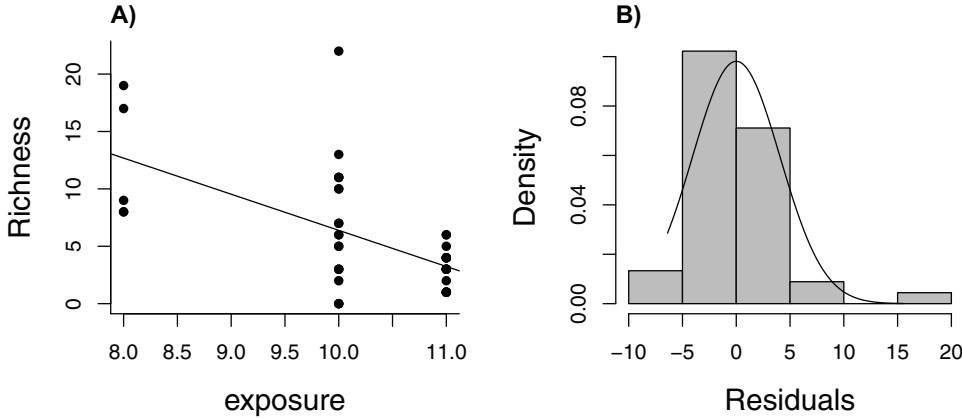

**Figure 6 Regression line with observed data (A) and histogram of residuals (B) for linear model relating species richness to the level of exposure of the beach measured at 45 stations in the Netherlands.**

the bootstrap to mimic the way data were collected (i.e., cluster sampling, with multiple observations from each of several beaches). In this case, we need to sample *beaches* with replacement, keeping all observations from a beach when selected (we will refer to this approach as a cluster-level bootstrap). Doing so allows us to evaluate how much our estimates of regression parameters would change if we were to collect another sample of observations from a different set of nine beaches. Our estimate of uncertainty would not require assumptions of Normality or constant variance. Further, we would only need to assume that data from different beaches are independent (not that observations from the same beach are independent). Although it would be nice to have a larger sample of beaches to represent the population, alternative analysis methods (e.g., models appropriate for clustered data) would face a similar challenge of having to estimate among-beach variability from nine beaches.

Conceptually, this example is no more difficult to understand than the fisheries slot limit example. We are trying to determine how much our regression parameter estimates would vary if we were able to collect another sample of observations using the same sampling effort (nine beaches, each with five stations). Implementation becomes slightly more challenging than the previous examples because we can no longer rely on the `resample` function in the `mosaic` package (it does not accommodate cluster-level bootstrapping, but see *Deen & De Rooij, 2019*, for an alternative using the `ClusterBootstrap` package). The bootstrap distributions of the regression coefficients in this example are not symmetric, so it is beneficial to consider alternative bootstrap confidence interval procedures. It is important to note that not all confidence intervals perform equally well. Normal- and percentile-based intervals are simple but less accurate on average than intervals that attempt to account for bias and skewness of the sampling distribution (e.g., the "BCa" interval illustrated in Appendix B); accuracy is measured in terms of an interval estimator's *coverage rate*, defined as the frequency with which the interval estimator includes the true population parameter. We include, as Supplemental Material, code necessary to implement the cluster-level bootstrap, demonstrating a range of

confidence interval procedures implemented using the `boot` package (*Canty & Ripley, 2019*; Appendix B).

## Case Study II: combining data sources: estimating uncertainty when the sampling distribution is unknown

As mentioned above, the Central Limit Theorem guarantees that the sampling distribution of many estimators, including all maximum likelihood estimators, will be Normally distributed as sample sizes become large (*Casella & Berger, 2002*). In practice, however, it may be difficult to derive an analytical expression for the standard error to use in conjunction with a Normal approximation. This is particularly true for estimation problems that require fitting multiple models to different response variables collected from the same or different sets of sample units. As an illustrative example, we consider data from *Zicus, Rave & Fieberg (2006)*, used to evaluate the relative cost-effectiveness of single- and double-cylinder nesting structures for mallard (*Anas platyrhynchos*) ducks. To measure duckling productivity, *Zicus, Rave & Fieberg (2006)* fit a linear regression model to duckling counts collected from 110 nesting structures over an 8-year period. To estimate costs, they fit an additional discrete-time survival model to quantify the probability that a structure would fall over and hence require a visit (with associated maintenance costs) for continued use. Both models included predictors capturing cylinder type (single or double) and size of the wetland holding the structure. *Zicus, Rave & Fieberg (2006)* combined the output from the two models and estimated, for a range of wetland sizes and for both cylinder types, the expected cost of maintaining the structure divided by the expected number of ducklings produced as a measure of cost-effectiveness. Estimating the standard error of this cost-effectiveness measure was complicated for several reasons: (1) the quantity of interest was a ratio and therefore not a linear function of model parameters, (2) there was a complex dependence structure resulting from repeated measures on the same set of structures, which were also shared across the two models; and (3) the residuals from the duckling model were not Normally distributed.

Fortunately, using a bootstrap to estimate a confidence interval for cost-effectiveness was relatively straightforward. We just generated new bootstrapped data sets, mimicking the original sampling design, and repeated the analysis using these new data sets. As illustrated in Appendix C, we: (1) resampled nest structures with replacement; (2) fit both models to the bootstrapped data sets; and (3) used the fitted models to estimate the expected number of ducklings, expected structure survival (and hence cost), and the ratio of expected cost to the expected number of ducklings for each cylinder type at different wetland sizes. This process allowed us to calculate confidence intervals for the cost effectiveness of both types of nest structures across a range of wetland sizes (Appendix C). We have used this same general approach elsewhere to quantify uncertainty in estimated growth rates from population models when parameters are estimated from multiple data sources (e.g., *Ellner & Fieberg, 2003*; *Fieberg et al., 2010*; *Lenarz et al., 2010*).

One important consideration in this example is that our estimator of cost-effectiveness, a ratio of expected means, may be biased. We can also use the bootstrap to estimate

Bias = $E[\hat{\theta}] - \theta$, where $E[\hat{\theta}]$ refers to the mean of the sampling distribution of $\hat{\theta}$, our estimator of the unknown parameter $\theta$. To understand how this works, consider that we use the bootstrap to mimic sampling from our population: the bootstrap samples relate to the original sample in much the same way as the original sample relates to the population (*Hall, 1988*; Fig. 2). Thus, to estimate bias, we can compare the mean of our bootstrap statistics, $\theta_b^*$, to the same statistic calculated using our original sample, $\hat{\theta}$, as though the sample were the true population. In Appendix C, we estimate the bias of our cost-effectiveness measure for both deployment types and across a range of wetland sizes, showing that the bias is small relative to the standard error.

## Case Study III: evaluating model-selection uncertainty

The bootstrap can be used to quantify model uncertainty in addition to sampling variation (*Buckland, Burnham & Augustin, 1997*; *Fieberg & Johnson, 2015*; *Harrell, 2016*). Consider, for example, the common situation in which a researcher desires to develop a predictive model by selecting a limited number of explanatory variables from some larger set (e.g., using AIC statistics associated with various sub-models). Several models may provide nearly equally good fits to the data, in which case we might expect a different model to come out as "best" each time we collect a new data set (where "best" is determined via backwards stepwise selection using AIC). We can explore whether this is the case by fitting the same set of candidate models to each of several bootstrapped data sets.

In this case study, we explore model-selection uncertainty arising from backwards stepwise selection using AIC (Appendix D). We begin by fitting a linear regression model to abundance data of longnose dace (*Rhinichthys cataractae*) collected from the Maryland Biological Stream Survey (downloaded from http://www.biostathandbook.com/multipleregression.html). Starting with six predictors (measurements of various stream attributes), we use the `stepAIC` function in the `MASS` package in R (*Venables & Ripley, 2013*) to sequentially eliminate predictors until we can no longer reduce AIC further by dropping any of the remaining predictors. Implementing this process resulted in a reduced model containing three variables: acreage drained by the stream, maximum depth, and $NO_3$. We can then use functions in the `rms` (regression modeling strategies) package in R (*Harrell, 2019*) to explore variability in the best model as determined by backwards stepwise selection. First, we refit the full model using `ols` (this function is equivalent to `lm`, but it stores additional information with the model fit). We then use the `validate` function to choose a best model by applying backwards selection to each of 100 bootstrapped data sets (i.e., `validate` fits a series of models, using backwards selection to choose the best model for each bootstrapped data set). Applying this process resulted in 31 different best models. The most frequently chosen model, selected in 24 of the 100 cases, included only acreage and $NO_3$. The model originally chosen using `stepAIC` rose to the top in only nine of the 100 bootstrapped data sets. One bootstrap sample resulted in a model with all six predictors, and five bootstrap samples led to a model with a single predictor (maximum depth in four instances and $NO_3$ in one instance). In addition to demonstrating the high degree of uncertainty associated with choosing a

best model, these bootstrap results could be used to calculate model-averaged predictions (using the frequency with which a model was selected to determine model weights; *Buckland, Burnham & Augustin, 1997*).

## DISCUSSION

Biologists need to understand, quantify, and communicate measures of effect sizes and their uncertainty. With frequentist statistics, we conceptualize and measure uncertainty by considering how statistics (e.g., means, proportions, regression coefficients) vary from sample to sample; yet, in reality, we typically have only one sample at our disposal. Resampling-based methods provide a natural way to understand foundational concepts related to uncertainty, including sampling distributions, standard errors, confidence intervals, and *p*-values. This understanding, in turn, makes it possible to develop custom analyses for a range of messy data scenarios using the same set of core concepts.

Consistent with the general call to provide estimates of effect sizes and their uncertainty (e.g., *Johnson, 1999*), our case studies emphasized applications of the bootstrap in this vein rather than resampling-based methods for conducting hypothesis tests. The bootstrap can be viewed as a Swiss-army knife of statistical tools, providing estimates of uncertainty for a wide range of problems; we just need to: (1) use our sample data to estimate the distribution of variable values in the population; (2) generate new data sets by repeatedly sampling from this approximated population; and (3) analyze each set of data in a consistent manner. To some readers, it may feel like bootstrapping is cheating because it seems to make up data. In the fisheries example, we had a total of 154 observations of fish length, but with a couple of lines of R code and a computer, we produced 10,000 such data sets. Something definitely seems fishy. In fact, the name bootstrapping derives from the phrase "to pull oneself up by one's bootstraps," a physical impossibility. Are we really getting something for nothing? No, we are not generating any new data. What bootstrapping does is explore the internal variability of a single data set. Formula-based estimates of uncertainty operate in a very similar manner but are available in only a few select cases where we can derive the sampling distribution from our underlying model assumptions or via the Central Limit Theorem. For example, the formula for the standard error of a sample mean, $SE = s/\sqrt{n}$ also uses a measure of internal variability (the sample standard deviation, *s*) to quantify variability among repeated samples. A mathematical justification for the bootstrap procedure was provided by its developer, *Efron (1979)*, extending work on the jackknife, a linear approximation to the bootstrap (*Quenouille, 1949*).

Important to note is that bootstrapping can be only as good as the original data. This means that the sample data must be representative of the population from which they were drawn, preferably randomly, and that the sample size should be adequate to capture the variation in the population. When generating new data sets, it is also important to mimic the way the original data were collected and to identify independent sample units for resampling. In our first case study, we resampled beaches rather than individual observations because we expected observations from the same beach to be correlated. Similarly, one can create "blocks" of observations close in time or space and treat these as

independent sample units when data are spatially or temporally correlated (*Chernick & LaBudde, 2011*). Although we focused on methods that rely on case resampling, which approximates the distribution of values in the population by making many copies of the sample data, parametric bootstrapping is also possible. In the latter case, we must assume the population values follow a particular parametric distribution (e.g., gamma, beta, etc.), use the sample data to estimate parameters of this distribution, and then use this parameterized distribution to simulate new data sets.

In addition to the examples presented in conjunction with our case studies, bootstrapping can be used to provide accurate estimates of model fit (e.g., $R^2$) and calibrate models (*Harrell, 2016*; Appendix D), to reduce prediction variance when using "greedy analysis methods" with many unknown parameters (e.g., random forests as a solution to the instability of regression trees; *Breiman, 2001*), and to quantify uncertainty across a diverse set of applications (*Davison & Hinkley, 1997*; *Manly, 2006*). We find the bootstrap particularly attractive for decision analysis because it can provide a distribution of possible outcomes across a set of potential management actions while accounting for model and parameter uncertainty (e.g., *Ellner & Fieberg, 2003*). Ecologists have a long history of using resampling-based methods to analyze multivariate response data using ordination methods (*ter Braak, 1990*; *Vendrig, Hemerik & ter Braak, 2017*; *Van den Brink & ter Braak, 1999*; *ter Braak & Smilauer, 2018*). The `vegan` and `permute` packages provide methods for restricted permutations (e.g., permuting observations within blocks for randomized complete block designs) that can be used with multivariate data (e.g., *Anderson & ter Braak, 2003*; *Oksanen et al., 2019*; *Simpson, 2019*); the `mvabund` package also provides methods for bootstrapping generalized linear models fit to multivariate response data (*Wang et al., 2019*; *Wang et al., 2012*). Another interesting example in biology is the use of the bootstrap to quantify the level of confidence associated with different clades on a phylogenetic tree (*Felsenstein, 1985*; *Efron, Halloran & Holmes, 1996*).

We acknowledge that fully model-based alternatives exist for addressing many of the challenges encountered in our case studies. For example, random effects could be used to account for potential correlation among observations from the same beach, and an appropriate count distribution (e.g., Poisson or negative binomial) could be used to model the counts rather than a Normal distribution. Hierarchical models with random effects "feel good" because they match the underlying structure of many ecological data sets. Further, mixed effect models offer significant advantages for modeling hierarchical data as these approaches allow one to consider both within- and between-cluster variability. For example, we could have allowed each beach to have its own intercept drawn from a Normal distribution. Rather than relax assumptions, however, this hierarchical approach would add even more assumptions (*Murtaugh, 2007*). Before trusting inference from the model, we would need to evaluate whether the Normal distribution was appropriate for describing variability among beaches. Adding random effects also implies that observations from the sample cluster (in this case beach) are *positively* correlated (*Fieberg et al., 2009*). Although this assumption is reasonable for these data, it may not be appropriate in other situations. For example, the lead author recently collaborated on a survival analysis of moose calves that included data from several twins

(*Severud et al., 2019*). The fate of twins may be positively correlated due to genetics, mother's health, and environmental effects, but it is also possible that losing a calf might increase maternal investment for, and the survival probability of, its twin. Using a cluster-level bootstrap allowed the fates of these individuals to be positively *or negatively correlated*, whereas using random effects would not (*Smith & Murray, 1984*).

A simple cluster-level bootstrap provided a reasonable solution to non-independence and non-Normal data in our first case study because we had balanced data (i.e., equal numbers of observations for each cluster). Similar to Generalized Estimating Equations that use a working-independence assumption, this approach may be sub-optimal when applied to unbalanced data and potentially problematic depending on the mechanisms causing variability in sample sizes among clusters (e.g., if the size of the cluster is in some way related to the response of interest; *Williamson, 2014*). Also of note, we were interested in the effect of a predictor variable, exposure, which did not vary within a cluster (i.e., exposure was constant across all measurements at a beach). Models that use random effects to model correlation offer substantial advantages when interest lies in predictors that vary both within and among clusters (*Muff, Held & Keller, 2016*). A parametric bootstrap (i.e., simulating from a fitted model) is always possible with random-effects models (e.g., using the `bootMer` function in the `lme4` package; *Bates et al., 2015*). This approach does not allow one to relax model assumptions, but could prove useful for estimating confidence intervals for functions of model parameters in situations where one is uncomfortable assuming the sampling distribution is Normal or where it is impossible to derive an appropriate standard error. Alternatively, *Warton, Thibaut & Wang (2017)* proposed bootstrapping probability integral transform (PIT-) residuals as a general method appropriate for non-Normal, and possibly clustered or multivariate, data. Model-based and resampling-based solutions to regression problems, particularly those involving dependent data (e.g., repeated measures, time series, spatial data), tend to be fairly complex. Therefore, it is important to consult with someone who has expertise in these areas and to recognize that statisticians may not agree on a best solution.

We occasionally hear from colleagues that it is hopeless to try to teach frequentist statistics to biologists—for example, it is impossible for non-statisticians to truly understand and interpret confidence intervals—and therefore, we should just teach Bayesian methods. Although replication plays a much lesser role in Bayesian statistics, we note that Bayesian $p$-values are frequently used to perform goodness-of-fit tests (*Kery, 2010*), and understanding the properties of estimators across repeated samples remains critically important for evaluating Bayesian estimators (*Rubin, 1984*; *Little, 2006*). In particular, demonstrating that Bayesian procedures are well calibrated (e.g., 95% credible intervals contain parameter values used to simulate data roughly 95% of the time) can help overcome the criticism that Bayesian methods rely on a subjective form of probability, with results dependent on one's chosen priors. This point cannot be overstated, particularly in today's highly polarized world in which individuals have strongly held prior beliefs that differ among groups.

In our own work, we take a pragmatic approach to data analysis and use a variety of tools, including frequentist and Bayesian model-based inference as well as

resampling-based methods. Yet, we find that resampling-based methods often provide an easier entry point to appropriate statistical analyses when consulting with less experienced and less-mathematically savvy users. We have also found resampling-based methods helpful for teaching foundational concepts in frequentist statistics and that undergraduate biology majors are able to adapt resampling-based methods to new problems. Thus, we see great opportunity in initiatives to use resampling-based methods to improve statistical thinking in the biological sciences (e.g., www.causeweb.org/STUB). In summary, we believe resampling-based methods should be used more frequently, both in practice and in the classroom.

## ACKNOWLEDGEMENTS

We thank S. Stapleton and the Government of Nunavut for sharing the polar bear data. We thank A. Gray, C. ter Braak, and N. Tintle for many helpful suggestions that greatly improved the manuscript.

### Funding

John R. Fieberg received partial salary support from the Minnesota Agricultural Experimental Station and the McKnight Foundation. The funders had no role in study design, data collection and analysis, decision to publish, or preparation of the manuscript.

### Grant Disclosures

The following grant information was disclosed by the authors:
Minnesota Agricultural Experimental Station.
McKnight Foundation.

### Competing Interests

The authors declare that they have no competing interests.

### Author Contributions

- John R. Fieberg conceived and designed the experiments, performed the experiments, analyzed the data, prepared figures and/or tables, authored or reviewed drafts of the paper, and approved the final draft.
- Kelsey Vitense conceived and designed the experiments, prepared figures and/or tables, authored or reviewed drafts of the paper, and approved the final draft.
- Douglas H. Johnson conceived and designed the experiments, authored or reviewed drafts of the paper, and approved the final draft.

### Data Availability

  All files (data, code, manuscript written using RMarkdown) needed to reproduce the article are available at https://bootstrapping4biologists.netlify.com/. The data are also

curated and archived with the Data Repository of the University of Minnesota:
DOI 10.13020/wn56-9y75.

## Supplemental Information

Supplemental information for this article can be found online at http://dx.doi.org/10.7717/
peerj.9089#supplemental-information.

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
