# Peer review of "Resampling-based methods for biologists"

_PeerJ, doi:10.7717/peerj.9089_

## Round 0.1 · original submission · Major Revisions

The reviewers have made some positive comments about your interesting manuscript, including its potential importance and the quality of your writing, and I agree with these. They have also raised some important issues that will need careful thought and potentially some significant reworking of your manuscript. All of their comments, and mine below, should be responded to point-by-point with either changes made to the manuscript or an argument for why no changes were made for each one.

Reviewer #1 makes a very good point about the use of code and I think this comes back to your target audience (I agree with Reviewer #2 that the manuscript is best seen as a “tutorial review”). The importance of this manuscript, for me at least, is introducing these concepts and providing some illustration of how they can be applied and I suspect that for some/many of the likely readers, R will seem a little arcane at first. I agree that if the code-based approach is taken, you need to be sure that potential readers are provided with enough foundations to appreciate the material (both code and output) you present, whether this is through providing more comments, a short tutorial to R, or some other approach. For example, the code between Lines 113 and 115 seem likely to confuse someone without some experience with R (particularly the second line). Their other comments are all very insightful and should also be extremely useful to you in revising your manuscript.

Reviewer #2 makes an also very good point about when these methods do not work well. I think that if the manuscript is to serve as a “tutorial review”, it is important that the reader doesn’t come away with the idea that this solves all of their modelling issues. As this reviewer notes, there are often important assumptions being made and a single best solution is not always accepted by all (bio)statisticians (e.g., resampling for mediation models, with random effects, etc.) and it seems crucial to me that the reader appreciates when they should approach a (bio)statistician or someone else with particular expertise in these methods. As a biostatistician who works with researchers, including from ecology, some of whom like to perform their own statistical analyses, this can be a challenging demarcation to make clear (and it will vary from non-statistician to non-statistician, as well as from (bio)statistician to (bio)statistician). I strongly suspect that a researcher like the ones you describe on Lines 40–43 will still struggle with some of these models when resampling is used. As with Reviewer #1, Reviewer #2’s other comments are all thought provoking and should be extremely useful to you when revising your manuscript. In particular, I think a zip file (either to be hosted on PeerJ or on your own website) containing all data and code together would be extremely beneficial to the reader.

Both reviewers have indicated that they are suggesting some of their own work and you are of course free to use or not use these. In this particular case, I think that their suggestions could be excellent additions to your manuscript and I hope that you will carefully consider each addition they recommend, but decisions around using these are entirely yours.

I will make some additional comments of my own to complement those from the reviews.

Line 8: I wonder if you might think it worthwhile to clarify that when you talk about normality, equal variance (and note that you use “constant variance” on Lines 29, 36, etc. and so might prefer to standardise this terminology), and independence, you’re referring to residuals (to approximate the distribution of the error term) and not to data (c.f. Line 29). I raise this especially as the sentence starts by talking about “Ecological data” and in my experience, this point is not always well appreciated by non-statisticians. If you agree, this could also be made clearer on Line 36.

Line 30: “and mostly rely ON analytical formulas”

Line 31: I wonder if this would be a good place to also mention non-parametric tests, correlation coefficients (including parametric and non-parametric), and even simple descriptive statistics (considering, for example, whether it is the sample mean or the sample median that is of actual interest is, in my experience, often neglected by researchers). These should all be included in an introductory course that extends to linear regression (I assume, but you should clarify, that this is the form of regression that you are intending on Line 28—the description here wouldn’t, for example, match logistic or Poisson regression—as per your reference on Line 33). This is the level I would associate with given your description on Lines 61–63, although those researchers who are entirely self-taught will also be covered by that. I completely agree that the traditional non-parametric approaches will not generalise to address more complicated situations.

Line 32: Given your comments about Chi-squared and F distributions above, and give my experiences with teaching such courses, I’d expect Chi-squared tests, one-way ANOVAs, and Kruskal–Wallis tests to also be covered. Perhaps it is more a matter that such courses often focus on univariable (i.e., one independent variable) and univariate (i.e., a single dependent variable) models rather than one or two sample problems? For a course that covers simple linear regression, I might, depending on the software being used, be tempted to include Huber-White standard errors and perhaps even clustered Huber-White standard errors as ways of relaxing some of the standard assumptions (along with unequal variance versions of the t-test and one-way ANOVA, and non-parametric tests for simpler models).

Line 44: Perhaps worth emphasising that the person performing the analyses does not have to perform this integration! The difference in interpretation of GLMMs and GEEs is, I think, often not well understood though and this is perhaps the point you were interested in here?

Line 49: I think you should consider if the challenges of appropriately preserving structure (including but not just clustering) in the data is sufficiently complex to warrant being a “main limitation”.

Line 111: Perhaps use, or at least note, the z-values of ±1.96 for the 95% CI here and/or on Line 127 (and perhaps again on Line 144 if that’s not too pedantic). Otherwise, I feel that some readers will be slightly confused when they look at other references.

Line 118: A reference or two on the CLT might be useful for some readers.

Line 235: Personally, I’d be much, much more cautious about a one-sided test here, only using this if a result in the opposite direction would absolutely not be commented on (even if it would otherwise have been statistically significant) rather than based on a belief that a result in the opposite direction was “unlikely”. Surprising findings, such as the negative correlations within clusters you mention later, shouldn’t be lost just because the researcher committed to a one-sample test.

Lines 252–253: In the presence of randomisation, confounding is not minimised but is rather completely eliminated. Confounding is a form of bias with a confounder being a variable that is associated with both the exposure and the outcome. With random assignment, any observed association between a “confounder” and the exposure can only be by chance and so cannot reflect any bias (systematic departure from the truth). Missing data mechanisms can induce such a bias, but in the absence of some such bias, confounding is not possible.

Lines 260–263: See comments from Reviewer #2. I think that this nicely illustrates the complexities of resampling-based approaches.

Lines 437–439: Of course, rather than modelling random effects, there are many covariance structures that could be imposed on the residuals, including allowing for negative covariances/correlations (although these are rather rarely the case in my experience).

Lines 445–452: I wonder if the segue into Bayesian methods is worthwhile here. While this is a subjective viewpoint, I’m not sure that a reader of the type you described back on Lines 61–63 will gain any real understanding from this brief aside.

I look forward to seeing the revised version of your manuscript.

·

Basic reporting

No comment

Experimental design

No comment

Validity of the findings

Overall this is a well-written paper on an important topic. However, there are some significant omissions in the presentation and lack of clarity on some of the methods/code that should be addressed.

Major issues

I think a significant limitation of the paper is staying so tied to R/Mosaic. I don’t find the R code intuitive enough (or well annotated enough). Other statistical applets freely available online (www.isi-stats.com/ and click through to the Applets page here: http://www.rossmanchance.com/ISIapplets.html , www.lock5stat.com and follow links to Statkey) have excellent visualizations that make explaining and understanding the concepts of resampling methods more accessible than reading R code. I think these are worth mentioning (or highlighting? Of course, these are methods I use personally and advocate for, so am somewhat biased here). Alternatively, I think that much more attention to explaining R functions, annotating and interpreting the code is needed.

Subsection on sampling distributions (pages 2 and 3). I find the presentation here confusing. I think that you need to provide (a) additional annotation of the R code and (b) additional annotation of the results. For example – a bit more explanation of what the functions means, and more statistical annotation (e.g., population mean vs. sampling means), that you are considering a ‘hypothetical’ situation in which you have all 164 values (the population) and are looking at what *could* have happened if you’d randomly sampled quadrats

Subsection on Central Limit Theorem (page 4). I think it would be helpful to reference back to the polar bears example here. Why not talk about how it ‘worked’ and using 2SE for the CI here, etc? Might also be worth pointing out that the discrete (and skewed) nature of the population in this case didn’t impact normality of the sampling distribution.
Bootstrap confidence intervals subsection (page 4). Maybe emphasize “many” a bit more to build intuition “many, many” or “infinite” . I would also point out that this ‘assumption’ that the sample is like the population (e.g., representative) is not a ‘new’ assumption that is being made. The term with replacement is used as if the reader knows what that means – I think I’d explain it, it’s a fairly technical term. I think this section doesn’t clearly enough state that you are sampling with replacement a sample the same size as the actual sample? Also, doesn’t clearly enough state that you compute the sample mean in that resampled sample. I wonder if a picture/visual would help here? Table/box that gives an overview of the method? Finally, of course, the beauty of the bootstrap is that you don’t have to use the mean! Median, other statistics, etc. are fine. You come back to this later, but not in a ‘single variable’ context – so maybe worth mentioning here?

R/code in this section is also similarly under-commented/explained to previous sections

Null distributions and p-values

It bothers me that the example you use here is not biological! Surely there is something more biological you could use?  Similarly, in a more ‘real’ context would you really want to do a one-sided test? Aren’t two-sided tests almost always conducted in practice?

Case studies

Could include something (or at least mention?) other experimental designs also? E.g., RCBD, or even just a novel statistic for some kind of case-control study (e.g., comparing genetic information between diseased/non-diseased folks)

Discussion

I think there are a number of additional points of discussion possible here
1. There are actually lots and lots of kinds of bootstraps which can be better/worse in various situations
2. You don’t really talk about permutation tests or when permutation tests are appropriate vs. bootstrap (e.g., it’s more about study design similarity than tests vs. CIs – you can use permutation test SEs to do CIs and bootstrap SEs to do tests)
3. You don’t talk about benefits to bringing this into classes with students, grad students, etc. (or efforts to do that: e.g., www.causeweb.org/STUB, among others).

Additional comments

Potential typo: line 85 “…making use…”

·

Basic reporting

This paper aims to help students/biologists to understand statistics by popularizing resampling (bootstrap and permutation) by providing background of the basic procedures and three case studies. The text is well-written, well-structured and easy to follow. Nevertheless, I have a number of issues with the resampling procedures and associated claims. See General and Details below.

I missed reference to (Manly 2007) [fourth edition is almost there; Manly & Navarro Alberto 2020, I suppose] which also aims at biologists.
I missed proper treatment of permutation tests and of permutation tests for particular regression coefficient(s) in linear models (Y-permutation using the Freedman & Lane approach, see (Anderson & Robinson 2001) and X-permutation, see (Dekker, Krackhardt & Snijders 2007) for DSP (double semi-partialling) with general discussion and more references. In the context of hierarchical designs, see (Anderson & ter Braak 2003) and the R package permute. See also under Issues.
For references, see below.

Experimental design

This is a tutorial review.

Validity of the findings

In places, the authors are too optimistic in my view. The proposed procedures do not always deliver what is claimed, as laid out in a number of issues under General comments.

Additional comments

This paper aims to help students/biologists to understand statistics by popularizing resampling (bootstrap and permutation) by providing background of the basic procedures and three case studies. As such, the paper is neither original research nor a review. I will treat as a tutorial review.
The text is well-written, well-structured and easy to follow. Nevertheless, I have a number of issues with the resampling procedures and associated claims. Some of the issues boil down to the statement that the bootstrap and resampling procedures that the authors explain/introduce do not work in the situations where the bootstrap and resampling procedures are really needed. The critical phrase in the paper is “If the bootstrap distribution is bell-shaped” [and, I add, unbiased] then simple procedures works and, I would add, arguably parametric procedures as well... [see issues 3-6]

Issues

1. My main issue is on line 260-263, which entails a permutation test on the effect of x2 in a linear regression of y on x1,x2 and x3. The procedure using the mosaic package is to shuffle the values of x2 across the sampling units “thus meeting the assumption of the null hypothesis” [line 263]. However, shuffling x2 does more than that. It not only nullifies the regression coefficient associated with x2 but also removes any correlation between x2 and (x1 and x3) in the data. Even, the approximate correctness of the procedure cannot be guaranteed if x2 is somewhat correlated to x1 and x3. There are two know solutions that lead to approximately valid permutation tests: Y-residual permutation, known as the Freedman-Lane procedure and X-residual permutation known as DSP. See (Dekker et al. 2007) for a relatively recent update and discussion on these procedures.
To help you a little on this. You have X-permutation after line 261. If in
do(10000)*{lm(y˜x1 + shuffle(x2) + x3, data=dat)}
x2 is replaced by the lm-residual of x2 on x1 and x3 and the lm gives access to the t-ratio of the regression coefficient of x2 (b2/se(b2)), then you are in business. One should look at the null distribution of the t-value of b2 [instead of at b2 itself] because one needs an (asymptotically) pivotal statistic for such tests (both DSP and Freedman and Lane).

2. In the section on Null distributions and p-values, code is given for a Monte Carlo permutation test. It looks to trivial and OK, but in detail there is something wrong: the resulting p-value could be 0 [OK, rarely so, with 10000 shuffles]. P-values should never be 0 (Phipson & Smyth 2010). See also (Hemerik & Goeman 2018)[difficult paper] and [easier] Box 1 of Appendix A4 in (ter Braak 2019) for how to correct this.


3. The introduction mentions the difficulty students may have with GLMMs and I share the concerns of the authors with this [see (Wang et al. 2012) and (ter Braak, Peres-Neto & Dray 2017) where specific resampling procedures are applied to GLM to make up for missing random terms, the need for which are shown in (ter Braak 2019) with accompanying permutation tests on interaction in a GLMM]. However, despite optimistic claims elsewhere and in this paper, there are, in my opinion, no generally valid bootstrap or other resampling procedures for GLMMs. The case study is discussed below in more detail; in brief, the proposed analysis might work for the data at hand but is not a general solution. Specific solutions have been proposed for significance testing in specific balanced hierarchical designs, see the R-package permute and (Anderson & ter Braak 2003). These solutions should be mentioned.

4. The abstract and introduction mention three common assumptions in traditional parametric statistics: Normality, equal variance and independence. The paper should make clear which assumptions are then avoided/not-needed, when proceeding to resampling procedures. Such explicit statement is missing. Note that violation of independence (pseudo-replication (Hurlbert 1984)) is arguably a bigger issue than deviation from normality. See the line transect and time series options in the permute package for avoiding the independence assumption.

5. The ‘95% rule’ (Box 1 and line is a misnomer in my view. It is the 2SE rule, aiming to give 95% confidence intervals.

6. I miss an easy rationale for the bootstrap: the bootstrap sample relates to the sample as the sample relates to the population (sample : population == bootstrap sample : sample) as the sample is equal to the population from which bootstrap samples are drawn (Hall 1988). This rationale is essential for bias correction and studentized bootstrapping intervals. Note that the naive bootstrapping intervals and the mean + 2bootstapSE intervals do not work where you really need the bootstrap.

7. Case study I. The data at hand are nested: 5 stations within each of 9 beaches and the response is a count. The authors propose a simple regression and make up for all arguments against the simple regression by a cluster-level bootstrap (without a bootstrap of units within clusters). The issue with nested data is that there are two regressions possible, one between-beaches and the other within-beaches. The authors choose to resample beaches; so they focus on between-beach variability. That is OK if the predictor (sea level) does not very within beaches (or only a little), but would be nonsensical if sea level varies mostly within beaches. The reader does not know what is the case (neither do I as I did not search for the data...). Mixed models and glmms combine these two regressions in a supposedly optimal way; the combination that lm makes depends on the numbers within each beach (here equal numbers, so perhaps more or less ok, but suboptimal). The current description of the cluster-level bootstrap does not work for data that are unbalanced. All of this needs to be discussed, perhaps in a supplement. Moreover, mixed models and glmm are such an important advance that the example needs it, and an appropriate bootstrap (also, bootstrap stations of beaches sampled) applied to the glmm. Note that lme4 has the bootMer function.
Moreover, the proposal does not solve the issue mentioned under linearity (the negative values).

An area where a sort of cluster level bootstrap (bootstrapping ‘id’s with a series of observations) is applied, is multivariate analysis/ordination in ecology; see R packages vegan (functions anova.rda for example) and permute, both modelled after my commercial package Canoco (ter Braak & Šmilauer 2018) (ter Braak 1990)[the latter document is open-access], for example (van den Brink & ter Braak 1999) (Vendrig, Hemerik & ter Braak 2017). A (many)glm version is in R package mvabund.


Details

Line 85 making use [of] the
L133 and Box 1 “we can make many copies of our sample data and use the resulting data set as an estimate of the whole population”. I did not understand that you also merged the copies to get a ‘big’ population. The only estimate of the population that we have is the sample; why make copies? If you would like to stay with the idea of the copies I would suggest: “we can make many copies of our sample data, merge them and their use the resulting data set as the estimated population”. Or perhaps mock population.
L 142 “If the bootstrap distribution is bell-shaped” is not even sufficient for the next to be true. There should be no bias either!
L274 Data and both R and Rmd scripts of Case Studies should, in my view, be supplied in a zip as Supplement, in addition to the current pdfs, to facilitate inspection, study and use by readers. I did not have those, and I did to try to reconstruct everything.
L287 “but for larger values of NAP we would predict Richness will eventually become negative, which is not possible.” You do not resolve this by resampling. A solution is to log transform (but see (Warton 2017) and (Warton et al. 2016)) and apply lme, because of issue 7 (Case study I). I do not see any objection to use GLM instead, and because of issue 7, GLMM with a full cluster bootstrap.

L381 As said in L388, they also need to communicate what is estimated and the estimate with uncertainty.

L391 I leave this statement with authors. I have a different opinion. Bootstraps for general GLMMs is an underdeveloped area, in particular because the semi-parametric bootstrap runs into the difficulty that random effects are the result of shrinkage estimators. See appendix A1 of ter Braak 2019. You may also wish to cite the PITtrap for count and binomial data (Warton, Thibaut & Wang 2017).

Box 1 Second bullet. Add: This alternative is termed the parametric bootstrap.

Appendices: Please add Conclusions: after a shown result. Now the unexperienced reader has to find out himself.

Appendix B. Please check for bias. Also, except for the quantile bootstrap, the scale on which is bootstrap estimate is formed matter. Here we have a ratio, so it would be wise considering the log and backtransform (as in the boot package).







Cajo ter Braak 29 October 2019

Anderson, M.J. & Robinson, J. (2001) Permutation tests for linear models. Australian & New Zealand Journal of Statistics, 43, 75-88.https://doi.org/10.1111/1467-842X.00156
Anderson, M.J. & ter Braak, C.J.F. (2003) Permutation tests for multi-factorial analysis of variance. Journal of statistical computation and simulation, 73, 85-113
Dekker, D., Krackhardt, D. & Snijders, T.A.B. (2007) Sensitivity of MRQAP Tests to Collinearity and Autocorrelation Conditions. Psychometrika, 72, 563-581.https://doi.org/10.1007/s11336-007-9016-1
Hall, P. (1988) Theoretical comparison of bootstrap confidence intervals (with discussion). Ann. Statist., 16, 927-981.https://projecteuclid.org/euclid.aos/1176350933
Hemerik, J. & Goeman, J. (2018) Exact testing with random permutations. Test, 27, 811-825.https://doi.org/10.1007/s11749-017-0571-1
Hurlbert, S.H. (1984) Pseudoreplication and the design of ecological field experiments. Ecological Monographs, 54, 187-211
Manly, B.F.J. (2007) Randomization, Bootstrap and Monte Carlo methods in biology, 3rd edition. Chapman and Hall, London
Phipson, B. & Smyth, G.K. (2010) Permutation P-values should never be zero: calculating exact P-values when permutations are randomly drawn. Statistical Applications in Genetics and Molecular Biology, 9, Article39.https://doi.org/10.2202/1544-6115.1585
ter Braak, C.J.F. (1990) Update notes: CANOCO version 3.1. Agricultural Mathematics Group. http://edepot.wur.nl/250652, Wageningen
ter Braak, C.J.F. (2019) New robust weighted averaging- and model-based methods for assessing trait-environment relationships. Methods in Ecology and Evolution, 0.https://doi.org/10.1111/2041-210X.13278
ter Braak, C.J.F., Peres-Neto, P. & Dray, S. (2017) A critical issue in model-based inference for studying trait-based community assembly and a solution. PeerJ, 5, e2885.https://doi.org/10.7717/peerj.2885
ter Braak, C.J.F. & Šmilauer, P. (2018) Canoco reference manual and user's guide: software for ordination (version 5.10). Microcomputer Power, Ithaca, USA
van den Brink, P.J. & ter Braak, C.J.F. (1999) Principal Response Curves: Analysis of time-dependent multivariate responses of a biological community to stress. Environmental Toxicology and Chemistry, 18, 138-148
Vendrig, N.J., Hemerik, L. & ter Braak, C.J.F. (2017) Response variable selection in principal response curves using permutation testing. Aquatic Ecology, 51, 131-143.http://dx.doi.org/10.1007/s10452-016-9604-1
Wang, Y., Naumann, U., Wright, S.T. & Warton, D.I. (2012) mvabund– an R package for model-based analysis of multivariate abundance data. Methods in Ecology and Evolution, 3, 471-474.http://dx.doi.org/10.1111/j.2041-210X.2012.00190.x
Warton, D.I. (2017) Why you cannot transform your way out of trouble for small counts. Biometrics,
Warton, D.I., Lyons, M., Stoklosa, J. & Ives, A.R. (2016) Three points to consider when choosing a LM or GLM test for count data. Methods in Ecology and Evolution, 7, 882-890.http://dx.doi.org/10.1111/2041-210X.12552
Warton, D.I., Thibaut, L. & Wang, Y.A. (2017) The PIT-trap—A “model-free” bootstrap procedure for inference about regression models with discrete, multivariate responses. Plos one, 12, e0181790.https://doi.org/10.1371/journal.pone.0181790

---

## Round 0.2 · Minor Revisions

Thank you for your responses and edits. The two original reviewers have looked at your revised manuscript. One has indicated no further suggestions and the other has made some suggestions that ought not to be too difficult to address.

I’ve made some relatively minor additional comments myself below and I would hope that if you can address (through revision or rebuttal) each of the comments from Reviewer #2 and myself, then this will be the final round of revisions for what I think should be a very useful addition to the literature. Please note that several of my comments are stylistic and you should see these as nothing more than suggestions for you to consider.

Lines 156–158: While you might be assuming the n=75 case here (from two lines above), it seems useful to remind the reader that the CLT provides a good approximation to the Normal for sufficiently large sample sizes, where “sufficiently” will depend on the Normality of the population distribution and appears to include n=75 in this particular instance. You already cover this point back on Lines 150–151, where you could add “irrespective of the population distribution”, and again in several subsequent parts of the manuscript, so I’m only suggesting you consider a gentle reminder to the reader of this point here.

Line 311: I wonder if adding “absolute” here might be useful for readers: “the probability of observing aN ABSOLUTE difference in sample means as large or larger”. Again, this is made clear in the following text, so this is just a suggestion for you to consider gently reminding readers here. Related to this, you could consider adding “two-tailed” before “p-value” on Line 313.

Line 324: Rather than “sure”, do you mean “confident”? (As you say earlier on Line 191.)

Line 324: Do you want to keep the decimal places consistent here, i.e. -38.9 and 3.6, or is the use of two significant figures for both your preference?

Line 353: This is a very minor point, but is the switch from 10,000 samples to 1,000 intentional at this point? This seems to be the solitary instance of 1,000 samples in the body of the manuscript.

Line 368: Do you mean “test A statistical hypothesis” or “test statistical hypothesEs” (I’m guessing the latter here)?

Line 390: Since “data” is plural and you refer to “data and R code” on the previous line, perhaps “is” here should be “are”?

Lines 483–510: I wasn’t sure if there was a pattern here as to when you quote “best” (Lines 488, 499, 503, and 504) and when you don’t quote it (Lines 501 and 508).

Line 525: This is purely stylistic, but given your punctuation elsewhere (e.g. the following line), I was expecting a comma after “To some readers” here.

Lines 535–536: I appreciate that you are not trying to describe the history here, but I wonder if adding “extending work on the jackknife, a linear approximation to the bootstrap.” or similar would be useful here, and might be a term that some readers have encountered. You could add a reference to Quenouille (1949) if you really wanted to here.

Line 537: I apologise for another stylistic comment, but is the ordering of words in “can be only as good as the original data” intended or did you want “can only BE as good as the original data” (the latter seems much more common to me, although neither seems wrong)?

Line 599: As a final stylistic comment, with a hyphen in the parenthetical abbreviation “(PIT-)” should there also be one after the “transform” immediately prior?

Figure 5: Do you want to elaborate on the “black horizontal line” as there are two of these in the figure? E.g. “Our observed sample statistic, (\bar{x}_starved - \bar{x}_fed = -18.3, is indicated by the LEFTMOST black horizontal line.”

·

Basic reporting

Good.

Experimental design

Good.

Validity of the findings

Good.

Additional comments

You have addressed my concerns.

·

Basic reporting

This paper aims to help students/biologists to understand statistics by popularizing resampling (bootstrap and permutation) by providing background of the basic procedures and three case studies. The text is well-written, well-structured and easy to follow. The revision led to a much improved paper; my previous major concerns have been taken care of (two more minor ones left). See General and Details below.

Experimental design

N/A

Validity of the findings

The paper reflects, in my view, the state-of-the-art on resampling as possible in a tutorial review. On two points, the main text or supplement could be improved as detailed below in Comments:
1. the null hypothesis testing using bootstrap vs permutation when variances differ.
2. adjustments when only a small number of permutations can be taken e.g. because each run takes long, or interest in very small p-values, for example in the context of multiple testing.
This would make the text quite balanced and complete.

Additional comments

The revision led that to a much improved paper; most of my previous concerns have been taken care of.

Details

L99 add perhaps: “, throughout the paper in the format with first the R code and then its output” [I suggest this as I did not understand this format, for example, that in p316 tally gives this kind of output automatically].

L115 On reading I liked the 75 to be mentioned earlier. “subset of size 75 from all quadrats” or more simply/shorter, and in agreement with line 116-117 , “a random subset of 75 quadrats”.

L135. Perhaps replace by “2 standard deviations is exact for 21 data points, 1.96 for many, many data points”

L320. Add “ given a p-value of 0.13” ?

L344 I found an older reference for residual X-permutation (please check). Please add in front of Dekker et al 2007: (Collins 1987): Collins MF. 1987. A permutation test for planar regression. Australian Journal of Statistics 29:303-308.

L593. “Models that use random effects to model correlation offer substantial advantages when interest lies in predictors that vary within a cluster (Muff et al., 2016).” This statement is strictly true for binary response, but not for normal or Poisson distributions. For the latter, one can simply add a factor ‘cluster’ to the LM/GLM. For the statement to be true I would change to: “when interest lies in predictors that vary both within and between clusters”.

L600 “Alternatively, Warton et al. (2017) proposed bootstrapping probability integral transform (PIT-) residuals as a general method appropriate for non-independent data.”. I do not think so. Change “non-independent” to “non-normal”.

L255-392 On Fig. 4 and the null hypothesis and bootstrap vs permutation. The variances in Fig.4 differ greatly between groups. I missed something on the topic, namely that the null distribution using permutation comes from the null hypothesis that the two distributions are equal. The null hypothesis is thus, strictly speaking, not solely on mu. It is the test statistic that makes that the test has power for particular alternatives. Here the test statistic has power against differences in means, but it is might have inflated type I error if the variances differ (and the true means happen to be equal). Compare the Behrens-Fisher problem and papers that show that differences in variance lead to liberal tests on the mean. This aspect works in favor of the bootstrap test in which only the mean is shifted. Please consider what you can communicate on this, in the main text or supplementary. Related to this: you make no difference between approximate and exact resampling tests; perhaps better to leave the paper in this respect as is.

In your answers to comment you wrote “We understand this point (i.e., the p-value should not be 0 since the original data represents a possible permutation and should be counted in the numerator and denominator; not doing so will cause the p-values to be off by approximately 1/10000).” I would prefer if this subtle point was mentions somewhere, best perhaps in a Supplement, e.g. in the supplement for the partial regression coefficient. Note that some models may take long to run so that it is not easy to do 10000 resamples, or interest is in very small p-values, e.g. in high-dimensional data context. With the adjustment of including the test statistic of the original data, the (Monte Carlo) test is exact in the long run, even for 19 permutation.



Cajo ter Braak 30 January 2020

---

## Round 0.3 · accepted · Accept

Thank you for your revisions, which addressed all of the outstanding queries. There are a few small editing points that I’ve picked up when re-reading your manuscript, but these are minor enough for me to leave them for you to address as you see appropriate in the proofing stage. Some of my comments below are entirely stylistic, some are very pedantic, and some depend on possible (mis)readings of the text that might be pushing plausibility a little too much and I’m happy leaving any changes that you make in response to these entirely up to you. Well done on writing what I’m sure will be a very useful, and often referenced, article.

My apologies for not mentioning this earlier, but I wonder if a footnote or parenthetical comment around Line 122 along the lines that “If the reader executes this code, their values are likely to be slightly different due to their random number seed being different from ours. This also highlights the importance of explicitly setting the random number seed if results are to be reproducible.” might not be both reassuring and helpful to readers (most statisticians I know have one, and only one, horror story about needing to reproduce results when they had inadvertently relied on the system clock to provide the random number seed!)

Line 165: There is no space after the plus-minus sign (“…use estimate ±2SE to…”), but on Line 188, there is (“…estimate ± 2BSE…”). The space also appears in Figure 2 (twice) so you may wish to add one to Line 165 for consistency.

Line 173: There is an extra comma after the second “many” in “For example, we can make many, many, copies of our sample data…”

Line 232: While the differences referred to here will probably all be numerically different, this is not guaranteed and you could delete “different” from “Above, we stored the 10,000 different differences in means in a variable named…” without loss (this depends, I think, on exactly how you interpret “different” and whether a clone by chance is indeed “different”).

Lines 264, 287, 315, and 317: The minus signs here seem overly long to me, as if they were em-dashes perhaps? It’s definitely longer than the minus sign on Line 299, for example. My apologies if my eyes are deceiving me here. Looking more carefully, the minus sign back on Line 139 looks like it might be an ordinary hyphen rather than a mathematical symbol. You might prefer to use $\pm 2$ for that instance in any case (as you appear to do on Lines 165 and 188, for example). Similarly, the minus sign on Line 141 looks like an ordinary hyphen. What I believe should be an en-dash (for a range of values) on Line 199 looks like it might also be an ordinary hyphen as well. There may be other instances of these, but I’ll leave it up to you how pedantic you are about your typography.

Line 279: The text here extends into the right margin. See also the output between Lines 297 and 298, and between Lines 397 and 398.

Line 370: Given your pattern of comma use, I would have expected one after “applications” in “For computationally intensive applications it may not be possible…”

Lines 411–412: I wonder if “However, the permutation-based test may also reject the null hypothesis if the population means are the same in the two groups, but the population variances differ.” will be entirely clear to all readers. All hypothesis tests could reject the null given that it is in fact true, irrespective of any assumptions, and we can only hope that this is at the rate described by alpha. Perhaps this sentence could emphasise that the rejection rate could differ from the nominal alpha?

Line 430: You say “…the distribution of residuals is right skewed and, thus, far from Normal…” but right skew in itself, i.e. without any qualification as to the degree of skew, would not justify “far from Normal” (i.e. slight right skew would be close to Normal rather than far from it). Perhaps, “…the distribution of residuals is strongly right skewed and, thus, far from Normal…” or “…the distribution of residuals is right skewed and, thus, non-Normal…” depending on how you interpret Figure 6B.

Line 447: There isn’t any issue with the terminology “among-beach variability”, but, in my experience, “between-beach variability” would be more usual, at least for the hyphenated version (along with “within-beach variability”, c.f. the usual expressions between-subject and within-subject). You use the more normal hyphenated forms on Line 595 (“…both within- and between-cluster variability.”) but then also use among (in a non-hyphenated form) on Line 617 (“…both within and among clusters…”). (You also use “among” in other places, but these were the two instances when it caught my attention.)

Lines 456–458: Percentile methods do attempt to address skew in the sampling distribution. I also wondered if the “…less accurate than…” should be qualified, e.g. “…less accurate on average than…” as bias correction can still worsen coverage.

Line 501: I think I understand why you’ve used theta-hat here, as this is the statistic being used, but this sentence seems also able to be read as referring to “…the true population…[which is] theta-hat”. rather than “…the true population…[estimated by] theta-hat”.

Line 521: You might be doing this intentionally, which is fine, but I wondered about whether the “3” here should be subscript. See also Lines 528 and 530.

Box 1: For the final point, I think an “as” (or similar) might be missing, i.e. “The p-value is the chance of obtaining a sample statistic as extreme AS (or more extreme than) the observed sample statistic, if the null hypothesis is true.”

References: Shouldn’t “pva” on Line 690 be “PVA”? Also for this title, normally you capitalise the first letter after a colon (e.g. Lines 685 and 687) but here you’ve left “effects” with a lower case “e”. Line 745 also has a lower case letter after the colon in an article title, as does Line 770. There might be other instances of this, but generally you do use a capital after a colon in the article title.